# Beyond Formulation: Contributions of Nanotechnology for Translation of Anticancer Natural Products into New Drugs

**DOI:** 10.3390/pharmaceutics14081722

**Published:** 2022-08-17

**Authors:** Rodrigo dos A. Miguel, Amanda S. Hirata, Paula C. Jimenez, Luciana B. Lopes, Leticia V. Costa-Lotufo

**Affiliations:** 1Institute of Biomedical Sciences, University of Sao Paulo, Sao Paulo 05508-000, Brazil; 2Institute of the Sea, Federal University of Sao Paulo, Santos 11070-100, Brazil; 3Department of Human Biology, University of Cape Town, Cape Town 7925, South Africa

**Keywords:** anticancer drugs, natural products, drug delivery, nanotechnology, paclitaxel, doxorubicin

## Abstract

Nature is the largest pharmacy in the world. Doxorubicin (DOX) and paclitaxel (PTX) are two examples of natural-product-derived drugs employed as first-line treatment of various cancer types due to their broad mechanisms of action. These drugs are marketed as conventional and nanotechnology-based formulations, which is quite curious since the research and development (R&D) course of nanoformulations are even more expensive and prone to failure than the conventional ones. Nonetheless, nanosystems are cost-effective and represent both novel and safer dosage forms with fewer side effects due to modification of pharmacokinetic properties and tissue targeting. In addition, nanotechnology-based drugs can contribute to dose modulation, reversion of multidrug resistance, and protection from degradation and early clearance; can influence the mechanism of action; and can enable drug administration by alternative routes and co-encapsulation of multiple active agents for combined chemotherapy. In this review, we discuss the contribution of nanotechnology as an enabling technology taking the clinical use of DOX and PTX as examples. We also present other nanoformulations approved for clinical practice containing different anticancer natural-product-derived drugs.

## 1. Introduction

Natural products have been explored since ancient times as a strategy for treatment and healing of various maladies. In this sense, nature is recognized as an important source of chemical entities with potential to be translated into new drugs [1]. Between 1981 and 2019, approximately 49.5% of all drugs approved for marketing were natural products or derivatives, disregarding only vaccines, biological macromolecules, and genuinely synthetic compounds. For cancer, this percentage is even higher: 64.9% are natural product-based drugs [2]. Two examples of substances originally derived from nature with anticancer properties that prospered in the translational process are doxorubicin (DOX) and paclitaxel (PTX). They are broadly used as first-line treatment for a variety of tumors, such as breast cancer, ovarian cancer, aggressive lymphomas, and other solid tumors [3,4]. Despite their success, the research and development (R&D) process of DOX and PTX into new drugs were especially challenging. Even after the earliest approval of the conventional formulations for both drugs, the side effects related to their poor selectivity and toxicity and the unfavorable physicochemical properties of PTX required new strategies to enable safer administration.

The use of these two drugs in therapeutics was further changed with the introduction of nanotechnology. By integrating concepts of chemistry, engineering, biology, and medicine, new nanocarriers can be developed to transport drugs through patient’s body with improved safety without compromising efficacy, among other goals [5,6]. According to the U.S. Food and Drug Administration (USFDA), nanotechnology-based goods “are products that contain or are manufactured using materials in the nanoscale range, as well as products that contain or are manufactured using certain materials that otherwise exhibit related dimension-dependent properties or phenomena”. In general, nanomaterial dimensions should range from 1 to 100 nm, but may reach 1000 nm if such product acquires distinct properties as a consequence of its dimension [7]. Some attributes conferred by nanotechnology were recently commented by our group using drugs in different stages of the R&D pipeline as examples, including those in preclinical and clinical studies, as well as approved medicines [8]. However, one question comes to mind when considering the applications of nanotechnology: is it worth investing in novel nanotechnology-based formulations for old drugs while conventional dosage forms are available? In other words, once development demands considerable time and resources, what are the actual rewards of nanotechnology for pharmacotherapy?

In this manuscript, we aim to examine the contributions of nanotechnology for clinical use of DOX and PTX in the treatment of cancer, highlighting its relevance in modulating pharmacokinetic properties and dosing, reversing multidrug resistance, protecting drugs from degradation or early activity, influencing the mechanism of action, and enabling administration by alternative routes and co-encapsulation with other drugs. Moreover, we present other possible contributions of nanotechnology under evaluation and other examples of anticancer drugs derived from natural products that exhibited similar challenges to DOX and PTX during the R&D process and, thus, may also benefit from nanotechnology-based strategies.

## 2. Doxorubicin (DOX) and Paclitaxel (PTX): Discovery, Mechanism of Action, and Conventional Formulations

### 2.1. Discovery of the Prototypes

In the 1940s, after the discovery of the antitumor activity of the antibiotic actinomycin A derived from the bacterium *Actinomyces antibioticus*, a special interest in the activity of this class of substances emerged. From 1959 onwards, a series of studies described the new species *Streptomyces peucetius* and the production of a potent antitumor antibiotic, daunorubicin, also known as rubidomycin or daunomycin. This discovery raised the hypothesis that structurally related compounds could originate new successful antitumor agents [9]. Then, in 1969, scientists modified a parental culture of *S. peucetius* with the mutagenic agent N-nitroso-N-methyl urethane, which derived the strain *S. peucetius* var. *caesius*, responsible for producing adriamycin, better known nowadays as doxorubicin (DOX). The first studies pointed out that DOX (14-hydroxydaunomycin), an analogue of daunorubicin, presented more favorable therapeutic index and a broader spectrum of antitumor activity [10].

Paclitaxel (PTX), formerly named taxol, was isolated and had its antitumor action described in 1967. Its complex chemical structure was fully elucidated in 1971, being described as the earliest taxane to have a potent antineoplastic activity. This substance was part of a screening program from the United States National Cancer Institution (NCI) in the early 1960s, in which they searched for new anticancer natural products and, as a result, described the activity of the crude extract from the bark of the western yew *Taxus brevifolia* [11]. The pathway for PTX development as an important therapeutic option for various cancer types has not been simple; PTX was overlooked for a long time due to, mainly, supply issues and unfavorable physicochemical properties, taking nearly 30 years for its complete development [12]. Figure 1 illustrates the timeline of PTX and DOX discovery and development, from the description of antitumor mechanism of action to the novel nanotechnology-based formulations.

### 2.2. Mechanism of Action

The earliest study involving the elucidation of the mechanism of action of anthracyclines started in the late 1960s using daunorubicin as model compound [13]. The mechanism of action by which DOX promotes cell cycle arrest and cell death consists of (i) intercalation of the aglycone portion of DOX between DNA base pairs, forming strong complexes with DNA and, consequently, interfering with both DNA and RNA synthesis; (ii) stabilization of the cleavage site and inhibition of the resealing site of the enzyme topoisomerase II, providing DNA break; and (iii) promotion of free radical-mediated oxidative damage to DNA in the presence of iron or by the action of redox enzymes that convert DOX (a quinone) in a semiquinone entity, which also impairs DNA and RNA synthesis [13,14,15]. Recently, scientists have described another mechanism for DOX: the stimulation of *de novo* synthesis of ceramide, resulting in nuclear translocation of CREB3L1, a transcription factor that activates the expression of various genes, including p21, a tumor suppressor gene [16]. DOX mechanism of action is illustrated in Figure 2A.

The mechanism of action of PTX was initially described in 1979, when the first reports pointed out that the compound acted as a microtubule-stabilizing agent [17]. Nowadays, it is known that this mechanism occurs by PTX binding to the N-terminal end of β-tubulin subunit of the microtubule, which promotes cell cycle arrest at M and G2 phases, since the microtubules are involved in mitosis [18,19]. Several other mechanisms have been proposed to explain the cytotoxic effects of PTX, such as (i) chromosome missegregation on multipolar spindles during mitosis [20]; (ii) interference in cell basic functions associated to the microtubules such as signaling, trafficking, and transporting [21,22]; (iii) activation of p53 [23]; (iv) overexpression of genes related to stress of the endoplasmic reticulum (ER), which provokes Ca^2+^ release; (v) increase in reactive oxygen species (ROS) resulting from mitochondrial damage [24]; and (vi) underexpression of Bcl-2 (anti-apoptotic protein) and overexpression of BAX (pro-apoptotic protein) [25]. These alterations are responsible for triggering mitochondrial apoptosis through disruption of mitochondrial membrane potential and the consequent release of cytochrome C into the cytoplasm, followed by cleavage of caspases [26]. More recently, PTX was also associated with modulation of immune response by reprogramming M2-polarized macrophages into an M1-like phenotype via TLR4 activation, increasing the NF-κB activity and the production of IL-12, as well as the ability of dendritic cells to induce CD8^+^ T-cell responses [27]. Figure 2B illustrates the mechanisms related to PTX anticancer action.

### 2.3. Conventional Formulations

In 1974, the USFDA approved the first DOX formulation, Adriamycin^®^, which consists of DOX hydrochloride solution for intravenous injection. This formulation was able to treat a large variety of leukemias, lymphomas, and metastatic solid and neural tumors and could also be used as an adjuvant chemotherapeutic agent for the treatment of breast cancer. As a single agent, the recommended dose for Adriamycin^®^ varied from 60 to 75 mg/m^2^ every 21 days and, in combination therapy, between 40 and 75 mg/m^2^ every 21 to 28 days (Table 1). The most common side effects associated with DOX are alopecia, nausea, vomiting, increased risk to develop secondary malignant neoplasms, and severe myelosuppression, which results in an increased risk of acquiring microbial infection [28]. Most of these effects occur because DOX can act in both tumor and healthy cells [15]. Beyond these effects, Adriamycin^®^ also promotes cardiotoxicity via a cumulative dose-dependent effect; in fact, it is so relevant that USFDA has established a maximum cumulative dose of 300–500 mg/m^2^ of Adriamycin^®^ as the recommended to reduce the risk of cardiotoxicity [28]. Nevertheless, the mechanism related to this side effect is not completely elucidated; what is known so far is that it may occur through generation of reactive oxygen species (ROS) [29].

The first conventional formulation of PTX, Taxol^®^, was USFDA-approved in 1992 for the treatment of ovarian cancer and, in 1994, it was indicated for breast cancer. Taxol^®^ consists of a nonaqueous solution composed of PTX, polyoxyethylated castor oil Cremophor^®^ EL, and dehydrated ethanol. This formulation is administered intravenously, and its regimen depends on the existence of a previous treatment with another antineoplastic agent. In general, the dose varies from 135 to 175 mg/m^2^ over 3 or 24 h of infusion every 3 weeks, with a maximum tolerated dose (MTD) of 240 mg/m^2^ (Table 1) [30]. Due to its non-tumoral selectivity, PTX promotes similar side effects as DOX. In addition, Taxol^®^ presents other problems related to the excipients, especially Cremophor^®^ EL. Administration of Taxol^®^ in dogs resulted in toxic effects, such as drop in blood pressure [31]. The first patients that received Taxol^®^ presented severe hypersensitivity reactions, and one of them died of anaphylactic shock [31]. For this reason, further clinical studies were blocked for 5 years until the pre-treatment with antihistamines and steroids and the prolongation of drug administration over a 24 h period were demonstrated to limit the incidence and severity of acute infusion reactions [31]. Additionally, Cremophor^®^ EL promotes leaching of plasticizers, such as di(2-ethylhexyl) phthalate (DEHP) from polyvinyl chloride (PVC) bags and infusion sets, which requires the preparation of Taxol^®^ to be carried out in non-DEHP infusion systems and the use of in-line filters for drug administration [32,33]. Other side effects are observed when administering Taxol^®^, such as neutropenia and prolonged peripheral neuropathy, characterized by axonal swelling and degeneration, vesicular degeneration, and demyelination [34,35].

## 3. Approved Nanotechnology-Based Formulations for Doxorubicin (DOX) and Paclitaxel (PTX)

The idea of nanocarriers and targeted delivery derives from the concept of the “magic bullet”, idealized by Paul Ehrlich (1854–1915; Nobel Laureate in Physiology or Medicine —1908) in the beginning of last century. He envisioned the use of devices capable of eradicating bacterial intruders or malignant cells without harming the human body. Several other concepts and discoveries—from periods long before nanotechnology was being discussed—have laid the foundation for nanotechnology-based products. Alec Banghan, for example, pioneered the description of the spontaneous self-assembly of phospholipids to form concentric membrane systems in the 1960s, which later became known as liposomes. First denominated “tiny fat bubbles”, it took decades before liposomes were developed into drug carriers [36]. Although liposomes are the most well-known type of nanocarrier, these certainly are not the only ones employed in nanomedicne: polymer–drug conjugates also stand out as successful platforms for drug delivery [37]. In the case of DOX and PTX, the need of a “magic bullet” that could reduce drug and/or formulation systemic toxicities without precluding efficacy motivated the development and approval of liposomal and polymer-based nanocarriers. In this section, we discuss the structural aspects of nanoformulations approved for DOX and PTX, also illustrated in Figure 3.

### 3.1. Nanoformulations Approved for DOX

Doxil^®^ (Figure 3; Table 1), from Johnson and Johnson (New Brunswick, NJ, USA), was the first nanoformulation of DOX and the earliest USFDA-approved nanosystem, in 1995. One year later, it was also approved by the European Medicines Agency (EMA) with the name Caelyx^®^. Nowadays, Doxil^®^ is approved for treatment of AIDS-related Kaposi’s sarcoma, multiple myeloma, and ovarian cancer, while Caelyx^®^ has the additional indication for breast cancer. Doxil^®^ is based on sub-100 nm pegylated liposomes loaded with DOX and composed of N-(carbonyl-methoxypolyethylene glycol 2000)-1,2-distearoyl-sn-glycero3-phosphoethanolamine sodium salt (PEG-PE), fully hydrogenated soy phosphatidylcholine (PC), and cholesterol [38]. Doxil^®^ is administered using intravenous infusion, ranging from 20 to 50 mg/m^2^ per dose depending on the type of cancer being treated [39,40]. The maximum tolerated dose (MTD) determined for Doxil^®^ was found in phase I clinical trials as 120 mg/m^2^, with grade 4 leukopenia and stomatitis being the dose-limiting factors [41].

Liposomes are vesicles formed by one or more concentric lipid bilayers, most often constituted of glycerophospholipids and/or cholesterol, entrapping an aqueous core. They can load both lipophilic and hydrophilic compounds. As DOX is a water-soluble drug, incorporation into the aqueous core of the liposome is expected. However, passive loading strategies, such as lipid hydration, significantly reduced the amount of DOX packed in the system due to the low volume of the aqueous central core of small liposomes, thus resulting in a lower concentration of DOX than the required for therapeutic effects [42,43]. To improve encapsulation, an active loading strategy was used, in which the drug was entrapped after the formation of the liposomes via the prior generation of a transmembrane ammonium sulfate gradient. This process, associated with the early development of liposomes and initiated in the 1980s, highlights the hindrances related to the formulation optimization and efficacy evaluation, since Doxil^®^ was only approved in the mid 1990s [43].

Myocet^®^ was the second liposome-based product containing DOX. Differing from Doxil^®^, this formulation is a non-pegylated liposome (Figure 3, Table 1) that was approved in Europe in 2000 as first-line treatment for metastatic breast cancer in combination with cyclophosphamide. The liposomes are composed of phosphatidylcholine and cholesterol and are loaded by a citric acid gradient, which follows similar principles as those of Doxil^®^, with vesicle size in the range of 150–250 nm [44]. Intravenous infusion and an initial dose of 60–75 mg/m^2^ are employed. During Phase I/II clinical trials, the MTD was determined in the range of 75–135 mg/m^2^ [45]. Similar to Doxil^®^, Myocet^®^ took approximately 11 years since the first description of the development of the nanocarrier to the approval of this formulation by EMA [46].

As Doxil^®^ patent expired in 2010, a generic version called Lipodox^®^ was developed by Sun Pharmaceutical Industries Ltd. (Mumbai, India) and approved by the USFDA in 2013. Preclinical studies with Lipodox^®^ presented equivalent physicochemical properties to those of Doxil^®^, such as similar morphologies and concentrations of drug, lipids and excipients [47]. Additionally, this formulation presented comparable in vitro antitumor activity, toxicity and pharmacokinetic profiles to the original formulation [48]. Moreover, two multicenter Phase I clinical trials demonstrated the bioequivalence of the formulations in terms of efficacy and safety [49]. Nevertheless, there are preclinical and clinical studies that contradict these data. A study performed using a human ovarian cancer orthotopic mouse model described a significant reduction in efficacy and intratumoral concentration of Lipodox^®^ in comparison to Doxil^®^, which may have been a result of drug distribution and uptake by tumor cells [50]. Furthermore, in a clinical setting performed by the same group, the overall response rate of patients treated with Lipodox^®^ was 4.3% compared to 18% to those treated with Doxil^®^, despite their similar toxicity profile [51].

### 3.2. Nanoformulations Approved for PTX

#### 3.2.1. Polymeric Nanoparticles

Abraxane^®^, also called nab-paclitaxel, was developed by American BioScience (USA) and was approved in 2005 by the USFDA and in 2008 by EMA [52]. This formulation consists of human serum albumin nanosuspension loaded with PTX (Figure 3, Table 1), in which particles have approximately 130 nm and are prepared by high-pressure homogenization. The production method consists of mixing PTX with albumin in an aqueous solvent and passing the system under a jet of high pressure [53]. Albumin is the most abundant protein in plasma (60%), responsible for the transport of various substances in the blood. Among the advantages that motivated the development of albumin nanoparticles are (i) the presence of two sites for drug interaction and non-covalent binding in the protein structure; (ii) non-toxicity, non-immunogenicity, and in vivo biotransformation to harmless products (amino acids); (iii) possibility of undergoing transcytosis through endothelial cells; (iv) accumulation in tumor due to overexpression of secreted protein acidic and rich in cysteine (SPARC), an extracellular matrix-associated protein involved in various biological processes; and (v) cellular uptake by receptor-mediated endocytosis [53,54].

Abraxane^®^ is approved at 260 mg/m^2^ for breast cancer treatment after failure of previous chemotherapy. In association with other drugs, non-small cell lung cancer (NSCLC) and pancreas adenocarcinoma are also among the indications for Abraxane^®^ treatment. The MTD determined for Abraxane was 300 mg/m^2^ [55]. Other combinations and cancer applications have been widely explored in clinical trials, such as association with gemcitabine, a pyrimidine nucleotide [56,57,58], and with atezolizumab, a monoclonal antibody against the programmed cell death-ligand 1 protein (PD-L1) [59,60], with good results in terms of patient outcome.

PICN (paclitaxel injection concentrate for nano-dispersion) is a formulation based on polymeric nanoparticles in the size of 100–150 nm (Figure 3), approved in India in 2014 for the treatment of breast cancer. This formulation is composed of polyvinylpyrrolidone (pVP), cholesterol sulfate, and caprylic acid, and is prepared using Nanotecton^®^ technology, developed by Sun Pharma Advanced Research Co., Ltd. (Mumbai, India). PICN is indicated at 260 mg/m^2^ and presents an MTD of 325 mg/m^2^ (Table 1) [61,62].

#### 3.2.2. Polymeric Micelles

Another formulation approved for PTX is Nanoxel^®^ (Figure 3, Table 1), a nanoformulation comprised of polymeric micelles developed by Dabur Pharma Ltd.(Ghaziabad, Uttar Pradesh, India) and approved in 2006 by Drug Controller General of India (DCGI) for treatment of breast and ovarian cancers, NSCLC, and AIDS-related Kaposi’s sarcoma. The main difference between polymeric micelles and polymeric nanoparticles relies on the fact that the first ones are nanosized molecules of core–shell structure that are formed by the self-association of amphiphilic block copolymers when they are added to an aqueous solvent, whereas the second are solid colloidal particles with a size in the range of 10–1000 nm, in which the drug can be entrapped or encapsulated within the carrier, physically adsorbed on the surface of the carrier, or chemically linked to the surface [62].

This nanoformulation consists of 80 nm polymeric micelles composed of self-assembled copolymers of N-isopropyl acrylamide (pNIPAM) and vinylpyrrolidone (pVP), which are biodegradable and amphiphilic. PTX is incorporated in the hydrophobic core and released by surface erosion. As polymeric micelles, these nanocarriers can (i) provide increased solubility; (ii) enhance drug stability; (iii) be metabolized to innocuous products; and (iv) control drug release rates [63,64]. Furthermore, polymeric micelles have the advantage of a reduced risk of microbial growth [26].

Genexol^®^-PM was developed by Samyang Corporation (Seoul, South Korea) and approved in 2007 in South Korea. This formulation (Figure 3, Table 1) also consists of polymeric micelles that range in size between 20 and 50 nm and is composed of an amphiphilic diblock copolymer of monomethoxy poly(ethylene glycol)-block-poly(D,L-lactide) (PEG-PLA) [65]. It is approved for treatment of breast cancer and NSCLC and it has also been employed in combination with other compounds, such as gemcitabine [66] and cisplatin [67]. Genexol^®^-PM is employed in the range of 300–390 mg/m^2^, which reaches the MTD of 300 mg/m^2^ [68]. It took approximately 12 years from the first publication related to the early development of polymeric micelles of Genexol^®^-PM to its approval in South Korea [69].

Another formulation consisting of polymeric micelles is Paclical^®^/Apealea^®^ (Figure 3), developed by Oasmia (Uppsala, Swetzerland) and approved by EMA in 2018 for the treatment of ovarian cancer. This formulation contains the XR17 micelle platform technology, which consists of isoforms of N-retinoyl-cysteic acid methyl esters. Paclical^®^ is a powder for solution, originating structures of 20–60 nm for infusion. It is employed at a dose of 250 mg/m^2^ for ovarian cancer treatment (Table 1) [26,70,71].

#### 3.2.3. Lipid-Based Formulations

As was the case for DOX, PTX was likewise incorporated into liposomes. Lipusu^®^ was developed by Luye Pharmaceutical Co. Ltd. (Shangai, China) and approved in China in 2003 by the State Food and Drug Administration (SFDA). Lipusu^®^ is composed of phosphatidylcholine and cholesterol (Figure 3) and employed at 175 mg/m^2^ for the treatment of ovarian and breast cancers, as well as for NSCLC (Table 1) [72].

In 2016, another lipid formulation was approved in South Korea for treatment of gastric cancer: Liporaxel^®^/DHP-107, an oral dosage form composed of monoolein, tricaprylin, and tween 80. This formulation is employed at 200 mg/m^2^ and possesses the highest MTD: 600 mg/m^2^ (Table 1) [26]. The biggest advantage of Liporaxel^®^ is the suitability of the oral route of administration. It is a semi-solid wax composed of an edible lipid and a USFDA-approved emulsifier that melts at 30 °C, facilitating swallowing. The formulation swells in the presence of aqueous fluids of the gastrointestinal tract, originating a mucoadhesive sponge phase (Figure 3), which consists of a disordered and less viscous nanostructured system, when compared to cubic phase. Cubic and sponge phases are mesophases—systems consisting of intermediate states between liquids and solids, conserving both fluidity and structural organization, respectively. More than enabling adhesion to gastrointestinal mucosa, other advantages are related to DHP-107, such as absorption independent of food intake or bile secretion. Nevertheless, formulation administration does not eliminate overexpression of efflux pumps such as P-glycoprotein (P-gp), neither cytochrome P450 (CYP) enzymes in small intestines, and liver after PTX repeated dosing [73,74].

## 4. Contribution of Nanotechnology for Doxorubicin (DOX) and Paclitaxel (PTX) Therapeutic Use

The contribution of nanotechnology to pharmacology goes beyond simply providing a new formulation for old drugs. Indeed, regulatory approval relies on the fact that the small size confers distinct properties to drugs compared to their conventional formulations and bulk counterparts. One of the most well-known properties associated with the nanorange is increased drug solubility, but modification of drug distribution in the body and pharmacokinetic properties, along with the possibility of obtaining aqueous dispersions of lipophilic drugs, may also be associated with nanoformulations. In this section, we focus on the main contributions of nanotechnology for DOX, PTX, and other anticancer drugs derived from natural products (Figure 4). As Doxil^®^ and Abraxane^®^ were the most studied nanoformulations of DOX and PTX, respectively, the contribution of these formulations to pharmacokinetics and pharmacodynamics of these drugs are discussed in more detail.

### 4.1. Novel and Less Toxic Formulation Vehicles

Many new chemical entities entering the R&D process present poor water solubility, providing several formulation and delivery challenges. A major problem is the lack of pharmaceutically acceptable hydrophobic vehicles considered safe for parenteral administration. In fact, several adverse effects are associated with hydrophobic vehicles and surfactants employed as solubilizing agents, as previously discussed for Cremophor^®^ EL from Taxol^®^ [32].

Several approaches were evaluated before settling on Cremophor^®^ EL plus ethanol for PTX dissolution, including the use of cosolvents, oil-in-water emulsions, and micellar solubilization. Very early in its clinical evaluation, a high incidence of acute hypersensitivity reactions was observed (reaching 25–30% in some studies), most of which were attributed to Cremophor^®^ EL [12]. Because of the problems associated with PTX administration, all nanoformulations approved for this drug are Cremophor^®^ EL-free. Abraxane^®^ consists of a lyophilized powder albumin-based nanoparticles that should be dispersed in aqueous buffers for administration. In the case of pVP, employed both in PICN and Nanoxel^®^, a non-toxic and biocompatible profile can be observed; this polymer, as well as pNIPAM, is also known by its stimuli responsiveness for pH and temperature, which confers distinct drug delivery properties [75]. For Genexol^®^-PM, the employment of PLA promotes not only the improvement of drug solubility through its interaction in the micelle core, but also confers biodegradability to the nanosystem, since the polymer can be hydrolysed into lactic acid monomers, which are degraded by Krebs cycle [76]. XR17 also demonstrated a low toxicity profile, since it is a retinoid derivative [70]. Lipusu^®^, all DOX formulations, Liporaxel^®^, and even PICN presents lipids in their composition, which display the major advantage of biodegradability by lipolysis, generally resulting in a low toxicity profile [42,77].

### 4.2. Reduction of Drug-Related Toxic Effects and Improvement of Safety Profile

Clinical studies demonstrated that Doxil^®^ and Myocet^®^ are comparable in efficacy to conventional DOX, but the liposomal formulations have an improved safety profile, considerably decreasing the risk of cardiotoxicity, myelosuppression, and alopecia [78,79,80,81,82].

In the case of PTX, Abraxane^®^, as a Cremophor^®^-free formulation, provided a safer administration, with lower risks of hypersensitivity reactions, neutropenia, and prolonged peripheral neuropathy [54,83]. Another advantage related to this nanoformulation is the shorter infusion time: while Taxol^®^ requires at least 3 h for its administration, Abraxane^®^ is administered in 30 min. Moreover, incorporation of PTX in albumin nanoparticles enabled the administration of a higher dose of the drug—260 mg/m^2^ in comparison to 135–175 mg/m^2^ of Taxol^®^ [32].

Despite the reduction of drug side effects mediated by nanocarriers in comparison to conventional drugs, the occurrence of undesirable events was not completely reversed, and other side effects may appear. In the case of Doxil^®^, for example, palmar–plantar erythrodysesthesia (PPE), or hand–foot syndrome, characterized by erythematous skin lesions on the palms of the hands and the soles of the feet, has been reported. It is associated with the accumulation of the liposomes in these areas because of the higher density of sweat glands and the thick stratum corneum [84]. Even though it can cause considerable discomfort for the patient and therapy change/interruption, it can be managed by dose modulation combined to local treatments and change of habits. Nevertheless, the overall safety of Doxil^®^ is higher than DOX itself due to less cardiotoxicity events [85].

Another adverse effect related to Doxil^®^ administration is the complement-activation-related pseudoallergy (CARPA), an infusion reaction that confers flushing and shortness of breath to patients, promoted by the formation of membrane attack complex in the lipid bilayer of the liposome, which is associated to the previous existence of anti-PEG antibodies in the patient, with the consequent release of DOX [86,87,88]. Nevertheless, more recently, scientists discovered that CARPA could be managed by administering DOX-free PEGylated liposomes (known as Doxebo) as a pre-treatment, which acts by inducing tachyphylaxis, when repeated doses of Doxebo reduces the chance of occurring CARPA [89].

Compared to Taxol^®^, Abraxane^®^ favored the occurrence of sensory neuropathy due to higher dose of PTX administered and its action in axon microtubules. However, differently from Taxol^®^, this sensory neuropathy is short and can also be managed by modulation of dose and infusion rates [32].

### 4.3. Protection from Premature Drug Activity and Alteration in the Distribution Profile

The first outline of Doxil^®^ was performed with non-PEGylated liposomes; however, it resulted in a quick clearance of the nanostructures by the reticuloendothelial system (RES) and in a rapid release of DOX from the nanocarriers in plasma, which resulted in cardiotoxicity and, consequently, turned incoherent the application of this nanotechnology [43]. A PEGylated lipid nanosystem was developed to delay capture of the vesicles by RES, as PEG diminishes the interaction between liposome and phagocytes and avoids the binding of opsonins, thus extending circulation time in plasma and reducing the apparent volume of distribution. Genexol^®^-PM also presents PEG in its composition [84,90]. Additionally, the presence of cholesterol in Doxil^®^ and in Myocet^®^ promoted a longer circulation time of the liposomes in plasma since it prevented the removal of the constituent phospholipids by high-density lipoproteins (HDL), a process known as lipoprotein-induced vesicle destabilization, which provokes the immediate release of the encapsulated drug [91,92]. Other factors that contributed to the slow rate of drug release in plasma were the aggregation state of DOX as a fibrous bundle and the pH gradient across the liposomal membrane, both factors related to the process of incorporation of DOX to the liposomes via generation of ammonium (Doxil^®^) or citrate (Myocet^®^) gradient [93].

Abraxane^®^, in turn, provided a higher plasma clearance and volume of distribution than Taxol^®^, which is an indicator of a rapid and broad distribution of PTX. In fact, it is known that Abraxane^®^ delivers 49% more PTX to tumors than Taxol^®^ [76]. One of the explanations for this phenomenon is the formation of micelles by the surfactant Cremophor^®^ EL after its administration, which provokes a rapid elimination of PTX through renal clearance because of its size and, at the same time, contributes to the systemic toxicities associated with Taxol^®^ because of complement activation and plasma premature drug release [33,94,95]. Moreover, since albumin presents a long biological half-life, the pharmacokinetic properties of PTX are improved in Abraxane^®^ and its elimination is much slower than Taxol^®^ [76].

### 4.4. Tumor Passive Targeting

As previously mentioned, Paul Ehrlich envisioned the concept of the “magic bullet” to describe drugs that act directly at their intended targets [96]. This concept served as a starting point for tissue targeting by nanocarriers.

In early clinical trials with Doxil^®^, it was observed that liposomes accumulated in the tumor microenvironment [97]. This phenomenon is expected with all other nanocarriers described here for intravenous injection: Lipodox^®^, Myocet^®^, Abraxane^®^ [35], PICN, Lipusu [98], Nanoxel^®^ [64], Genexol^®^-PM [99], and Paclical^®^. This accumulation was later described to be a result of the enhanced permeability and retention (EPR) effect, which is a consequence of high permeability of blood vessels and compromised lymphatic drainage from tumor [42]. Moreover, in general, healthy tissues have lower permeability because of tight junctions. Therefore, entrapping a drug into a nanocarrier precludes its accumulation at healthy tissues even when intravenously administered, preventing several side effects promoted by poor drug selectivity [100,101].

The mechanisms of Abraxane^®^-mediated delivery are more peculiar. After its injection, the nanoparticles dissociate in the bloodstream, forming albumin–PTX complexes that are similar in nature to other drugs that have affinity for albumin [102]. These complexes tend to accumulate in tumors not only because of the EPR effect, but also because of transcytosis from blood vessels due to the binding of albumin–PTX to endothelial receptors. This mechanism is most likely mediated by albumin binding to glycoprotein 60 endothelial receptor (gp60), which initiates the formation of an endosome that, after crossing the cytoplasm, will fuse to other regions of the membrane of the endothelial cell, transporting albumin from the plasma to the tumor [35,103,104]. The mechanisms of drug delivery of Doxil^®^ and Abraxane^®^ are illustrated in Figure 5.

### 4.5. Distinct Routes of Administration

Despite the proposed mechanisms for nanocarrier accumulation in the tumor microenvironment after intravenous administration, distinct routes can be explored with nanotechnology. One example is Liporaxel^®^, an oral dosage form of PTX, which forms a mucoadhesive sponge in the stomach and in the upper intestine, enabling drug absorption [74]. The relevance of this formulation for PTX is more than improving its solubility: the presence of monoolein as a gelation agent and tricaprylin and tween 80 as viscosity reduction agents promoted the balance between facilitating the swallowing of the formulation and its adhesion to the gastrointestinal mucosa [105]. Besides intravenous and oral administration, nanotechnology was also approved by the USFDA for intrathecal administration of cytarabine encapsulated in multivesicular liposomes (DepoCyt^®^) for the treatment of lymphomatous meningitis [106]. Thus, it demonstrates that exploration of distinct routes for drug administration may also provide tumor targeting.

### 4.6. Influence on Drug Release and Mechanisms of Uptake

There are three distinct mechanisms proposed for releasing of DOX from the liposomes in the tumor site after accumulation. The first is related to the fact that DOX releasing promotes the opposite process described for loading: as the tumor cell secretes ammonia as a metabolite resulting from glutaminolysis, it causes a change in the chemical balance related to DOX precipitation inside the liposome, promoting formation of ammonium sulfate and releasing DOX [107]. A similar mechanism may occur with Myocet^®^ through the generation of a citrate gradient. Another factor that may contribute to DOX release, described for Myocet^®^, is the increased phospholipase activity observed in some tumor types, which may degrade the lipid bilayer of liposomes and consequently release the drug [93]. Additionally, the cellular uptake of liposomes and its lysosomal processing may be also related to drug release [108].

Tumor accumulation of Abraxane^®^ is favored by the overexpression of the secreted protein acidic and rich in cysteine (SPARC) in the membrane of the tumor cell, associated with albumin arrest in the microenvironment [109]. Albumin–PTX complexes might release the drug following the same pathway that the endogenous albumin is used as energy source by the tumor cell: endocytosis following albumin binding to receptors, which facilitates interaction of the drug with its therapeutic target [102,110]. However, it is not clear as to how this mechanism occurs for Abraxane^®^. Recently, the expression of caveolin-1, an important protein related to endocytosis and overexpressed in tumor cells, was associated with albumin–PTX sensitivity in in vitro models, suggesting that this protein might participate in the cell uptake of nanoparticles [111]. Doxil^®^ and Abraxane^®^ mechanisms for drug release are illustrated in Figure 5. In the case of polymeric micelles, other mechanisms play important roles in PTX release: the surface erosion of the hydrophilic core of the micelle and the pH sensibility of the nanocarrier (more specifically for Nanoxel^®^). After being uptaken by the cell, the micelle undergoes lysosomal processing, releases the drug, and finally enables PTX activity [64].

### 4.7. Reversion of Tumor Resistance to Chemotherapy

The generic version of Doxil^®^, Lipodox^®^, demonstrated inhibition of P-glycoprotein (Pgp) in a model of drug-resistant colon cancer cells (HT29-dx). Pgp is an efflux pump belonging to the ATP-binding cassette (ABC) superfamily of membrane proteins that shuttle various substrates across cell membranes using energy from ATP hydrolysis. Overexpressed in various cell lines, Pgp presents as substrates various anticancer drugs, such as DOX and PTX, which promote a considerable reduction of drug intracellular concentration and efficacy, resulting in tumor resistance and relapse. There are other transporters in the ABC superfamily, and a considerable amount of them contribute to tumor multidrug resistance, including the multidrug resistance proteins (MRPs/ABCCs) and breast cancer resistance protein (BCRP/ABCG2) [112,113]. Several mechanisms have been proposed to explain the ability of nanocarriers to help overcoming efflux transporter-mediated resistance. The inhibition of Pgp by Lipodox^®^, for example, involves two main mechanisms: (i) an alteration in the composition of the lipid rafts in resistant cells by the fusion of the components of the liposomes with the cell membrane, which, when affecting lipids surrounding Pgp, interferes with transport, and (ii) direct interaction with Pgp, promoting conformational alterations that impairs ATPase activity of efflux pump and transport [113].

Other reports demonstrated the influence of the nanocarrier components in transporter activity. Cetyltrimethylammonium bromide (a cationic surfactant) and Cremophor^®^ EL demonstrated a modulatory activity on Pgp when administered with DOX in highly resistant glioma cells, demonstrating a reduction of the half-maximal inhibitory concentration (IC_50_) in comparison to DOX treatment alone when dissolved in solutions (up to sevenfold lower) and as excipients of nanoparticles that encapsulated the drug (up to 4.7-fold lower) [114]. Moreover, solid lipid nanoparticles containing the surfactant Brij 78 promoted a reduction of DOX and PTX efflux via reversion of P-gp activity and ATP depletion [115,116]. The ability of nanocarriers to increase cell internalization is valuable to overcome Pgp transport.

### 4.8. Influence on the Mechanism of Action

Despite the large number of studies demonstrating the benefits of nanocarriers and their ability to improve efficacy, very little is known about their effects on drugs’ pharmacodynamics. Only few studies to date have been proposed to understand whether the nanocarriers modify a drug’s pharmacodynamics, enhance a known effect, or induce new mechanisms of action.

A recent study suggested that the higher efficiency of Abraxane^®^ compared to Taxol^®^, in a lung cancer cell line (A549), could be also related to the underexpression of glucosamine 6-phosphate N-acetyltransferase 1 (GNA1) within the biosynthesis pathway of uridine diphosphate-N-acetylglucosamine, which is essential for N-linked glycosylation and cell growth [117]. This effect was not observed for Taxol^®^. It has also been proposed that the nanoformulation promoted a superior reduction of cancer stem-cells, which are related to a higher rate of metastatic, resistant, and recurrent tumors, while the conventional dosage form promoted an increase in the number of this cell population [118].

Another study, with formulations in the early development stage, demonstrated that DOX can promote distinct mechanisms of cell death depending on the nanocarrier. This study describes the development of hexosomes and cubosomes, which were functionalized with folic acid, enabling tumor active targeting. One of the results of this study points out that the cubosomes with DOX promoted necrosis, while hexosomes with the drug triggered cell death by apoptosis, highlighting the importance of the nanocarrier design for the desired effect [119].

## 5. Other Possible Contributions of Nanotechnology for DOX and PTX

Several other nanoformulations for DOX and PTX are found in earlier stages of the R&D process. Although they are not the focus of this review, some of their contributions are noteworthy and will be presented herein.

### 5.1. Drug Release by Thermal Stimuli

Some nanocarriers, depending on excipient composition, may present distinct properties depending on temperature. One example is aforementioned: Liporaxel^®^, which melts at body temperature and facilitates oral administration. Nonetheless, drug release may also be affected by temperature. ThermoDox^®^, which already has a Phase III clinical trial completed [120], consists of liposomes that share similar properties as Doxil^®^, such as a minor chance of cardiotoxicity because of drug entrapment and tumor targeting by EPR effect and reduced clearance. These liposomes are also composed of dipalmitoyl PC, PE, and PEG as Doxil^®^, but they also present another PC derivative, a single-chain lyso-lipid: mono-stearoyl-phosphatidylcholine (MSPC), that presents a melting temperature (T_m_) of 39 °C. For this reason, when ThermoDox^®^ is administered and the nanocarriers reach a previously heated tumor (with mild hyperthermia of 41 °C), the drug is released in front of lipid melting and may exert its therapeutic effects [121].

### 5.2. Tumor Active Targeting

Along with passive targeting by EPR, nowadays, there are a range of papers evaluating tumor targeting by functionalization of the nanocarriers by conjugating them with antibodies, peptides, and growth factors [122]. Since nanocarriers present a high surface area-to-volume ratio, multiple bindings are possible; thus, many ligands may be explored. The phenomenon promoted by nanocarrier coating is called active targeting. One formulation based in active targeting achieved Phase II clinical trials: MM-302, consisting of PEGylated liposomes coated with anti-HER2 (human epidermal growth factor receptor 2), for breast cancer treatment. However, a clinical trial that compared the benefit of this formulation combined with trastuzumab over other “chemotherapeutics of physician’s choice” plus trastuzumab failed to demonstrate superior efficacy [123]. Although functionalization does not seem to enhance tumor accumulation of cytotoxic drugs in the target site, since it relies more on EPR effect, it increases tumor specificity and cell uptake [122]. Another example of ligand is lactoferrin (Lf). Lf is related to the superfamily of iron-binding glycoproteins (transferrins) and their receptors are usually overexpressed in tumor cells, since their metabolic activity is higher [124]. Lf-decorated nanocarriers may promote a rapid internalization into cancer cells and increase the sensitivity of resistant tumors to the action of DOX, overcoming chemo-resistance, and increment the expression of cytokines TNF-α and IFN-γ [125]. In another study using PTX, Lf could be also co-functionalized with another peptide (tLyP-1), which increased penetration across the blood–brain barrier/blood–brain tumor barrier [126]. Moreover, as mentioned in the previous section, nanocarriers may be also decorated with folate, since tumors cells overexpress folate receptors, which augments cell uptake [119]. Functionalization techniques are increasing every day and various ligands may find application.

### 5.3. Increase in Solubility

More than encapsulating a poor soluble drug and, thus, facilitating its dispersion in aqueous-based vehicles through increasing its apparent solubility, nanotechnology may actually increase drug solubility. Since dissolution is a surface phenomenon, increasing the particle specific surface area (surface area-to-mass ratio) by reducing size increases dissolution. For this reason, nanonization is one strategy that may be employed, since it propitiates an increase in both dissolution rate and in saturation solubility, which provides a bigger concentration gradient in biological media such as the gastrointestinal lumen [127]. In a study performed with PTX, drug accumulation at the tumor was greater and longer with nanocrystals intravenously administered when compared to Taxol^®^, and the nanoformulation was less toxic [128]. Nonetheless, there are no nanocrystals in clinical trials for cancer treatment [123]. It is important to highlight that ≈40% of new chemical entities are poorly soluble in water; in addition to formulation challenges, discussed in the previous session, drugs with low aqueous solubility often present low dissolution rate in aqueous biological fluids and, thus, low bioavailability. For this reason, nanotechnology is regarded as one of the most explored techniques for circumventing poor solubility.

### 5.4. Co-Encapsulation of Drugs

Since most nanocarriers present hydrophilic and hydrophobic moieties, drugs with distinct physicochemical properties can be co-encapsulated to enable the modulation of multiple signaling pathways, improve efficacy, and reduce the dose compared to the use of single compounds. In a study, DOX was co-encapsulated in a liposomal system with curcumin, a hydrophobic substance obtained from *Curcuma longa*, which provided modulation of DOX biodistribution, reducing its adverse reactions and improving efficacy. This can be explained by the fact that curcumin interferes with DOX redox processes and inhibits Pgp [129]. The inhibition of Pgp is not restricted to intravenous administration; in another study, PTX was co-encapsulated with elacridar, a Pgp inhibitor, resulting in a greater skin localization of PTX, since Pgp is associated with drug transdermal absorption. This strategy could optimize local treatment with PTX and minimize systemic adverse effects [130]. In another research, two hydrophobic drugs (PTX and C6 ceramide, also cytotoxic) with limited penetration across the skin were co-encapsulated in nanoemulsions, which enhanced cutaneous transport and potentialized cytotoxicity. The use of smaller doses of PTX could eventually reduce toxicity and avoid drug precipitation during formulation process [131]. Another example of co-encapsulation is explored in further sections: Vyxeos^®^, which presents cytarabin and daunorubicin in order to promote combined therapy with complementary mechanisms of action [132].

## 6. Cost–Benefits

Considering that nanotechnology does not completely eliminate adverse effects, one might argue whether nanoformulations are really worthy in comparison to conventional dosage forms considering their generally higher cost. Thus, one point that should be discussed here is the cost–benefit of nanotechnology-based products.

According to Drugs.com [133], nanotechnology-based products have higher prices: Doxil^®^ and Lipo-Dox^®^ are priced 11- to 12-fold higher than Adriamycin^®^, whereas Abraxane^®^ is nearly 86 times more expensive than Taxol^®^. In spite of the higher costs, from a pharmacoeconomic point of view, nanomedicine can be more cost-effective than conventional formulations, since it (i) can reduce costs associated with hospitalization, medical devices, and monitoring; (ii) might decrease the risk of nosocomial infections and serious side effects; (iii) might give greater chances of remission; (iv) allow patients to return to professional life faster, contributing with the economy of their country; and (v) can avoid immeasurable costs related to patient’s quality of life, such as pain, suffering, and anxiety [134].

In the case of DOX, the costs of Adriamycin^®^ and the liposomal formulations have not been compared yet. Nevertheless, pharmacoeconomic analysis of data from clinical trials were performed by comparing Doxil^®^ with other liposomal formulations and other chemotherapeutic agents. One of these studies was performed in patients with Kaposi’s sarcoma and determined that, despite a higher total cost, Doxil^®^ is cost-effective when compared to liposomal daunorubicin, since this formulation demands a ≈2.2-fold greater expenditure to achieve a response with a treatment. Additionally, when compared to topotecan, Doxil^®^ revealed lower overall treatment costs in patients with ovarian cancer because it was administered less frequently and required fewer interventions for toxicity [135].

In a study performed in Spain in patients with metastatic breast cancer in which treatment with anthracyclines failed or was not indicated, the cost-effectiveness of Abraxane^®^ was compared with Taxol^®^. This study evaluated parameters such as life of years gained (LYG); quality-adjusted life of years (QALY) gained; and incremental cost effectiveness ratio (ICER), which is a quotient of the differences in costs and effectiveness (in function of LYG and QALY) of the nanoparticulate formulation in comparison to the solvent-based one. When the formulations were administered every 3 weeks, the mean LYG were 1.44 for Abraxane^®^ and 1.17 for Taxol^®^. Despite the higher total cost of Abraxane^®^ (EUR 16,447) in comparison to Taxol^®^ (EUR 13,509) administered in the same regimen, the ICER was EUR 11,084 per LYG and EUR 17,808 per QALY, which indicates the cost-effectiveness of the albumin formulation. Moreover, when compared to the weekly injections of Taxol^®^ regimen, Abraxane^®^, injected every 3 weeks, demonstrated a mean saving of EUR 711 per patient in comparison to the conventional dosage form [136].

## 7. Other Approved Natural Anticancer Nanodrugs

Natural products are well known for their complex structures and high molecular masses [137]. With greater chemical complexity, hindrances associated with drug delivery, as already mentioned for DOX and PTX, arise. In this section, nanoformulations that incorporate other natural-product-derived drugs are discussed as further examples of the nanotechnology contribution to treatment. Figure 6 illustrates the year of approval, the name of the formulations, and the complexity of drugs.

### 7.1. DaunoXome^®^

In the same year of Doxil^®^’s approval (1995), the USFDA approved DaunoXome^®^, a daunorubicin citrate liposomal formulation for the treatment of HIV-associated Kaposi’s sarcoma. DaunoXome^®^ is comparable in overall effectiveness and safety to the standard combination-drug therapy for advanced cases of this type of cancer, which involves the administration of DOX, bleomycin, and vincristine. Moreover, DaunoXome^®^ promoted fewer side effects than the conventional chemotherapy with daunorubicin, since liposomal drugs tend to target both Kaposi’s sarcoma lesions and tumors [138]. Therefore, daunorubicin and DOX present similarities not only related to chemical structure, mechanism of action, and side effects, but also considering the employment and features of their nanoformulations.

### 7.2. DepoCyt^®^

In 1996, the USFDA approved DepoCyt^®^, a multivesicular liposomal formulation containing cytarabine for treatment of lymphomatous meningitis. Cytarabine is a cytotoxic drug that was synthesized on the basis of the discovery of C-nucleosides produced by the Caribbean sponge *Tectitethya crypta* [139]. This drug is an analogue of deoxycytidine and belongs to the class of antimetabolites that inhibits DNA polymerase activity and DNA repair [140]. The need to develop a nanoformulation for cytarabine derives from the fact that this drug acts specifically in the S phase of the cell cycle and, for this reason, the optimal antitumor activity occurs when cancer cells are exposed to low or moderate concentrations of drug over an extended period. This way, a greater proportion of the cells would have passed through mitosis, which requires repetitive dosing or continuous infusion schedules, since cytarabine also presents a short half-life. Moreover, the systemic chemotherapy of lymphomatous meningitis is limited by the poor penetration of the drug across the blood–brain barrier, which requires direct intrathecal administration. DepoCyt^®^, in turn, prolongs cytarabine half-life, allowing less frequent administrations and, in spite of being a liposome, it must still be administered by the intrathecal route, which enables drug targeting [106].

### 7.3. MEPACT^®^

MEPACT^®^ or mifamurtide is an EMA-approved (2009) nanosystem for the treatment of osteosarcoma, consisting of liposomal muramyl tripeptide phosphatidylethanolamine (MTP-PE), a synthetic analogue based on muramyl dipeptide, a constituent of the Gram-positive and Gram-negative bacterial cell wall. Mifamurtide acts as a modulator of the innate immunity through the potent activation of macrophages and monocytes in the tumor microenvironment, after intravenous injection and accumulation in tumor site. The addition of mifamurtide to standard chemotherapy improves the overall survival from 70% to 78% and results in a reduction of 33% in the risk of death from osteosarcoma [141]. Thus, nanotechnology enables not only drug but also antigen delivery for immunostimulation.

### 7.4. Marqibo^®^

Marqibo^®^ is a nanoformulation that consists of vincristine sulfate liposomes and was approved in 2012 by the USFDA [142]. Vincristine was initially discovered in a screening program for the investigation of the potential antidiabetic properties of extracts from the white- or pink-flowered periwinkle plant (*Catharanthus roseus*). This drug binds to tubulin and inhibits microtubule polymerization, which consequently provokes metaphase arrest and apoptotic cell death. By this mechanism, vincristine potently inhibits leukocyte production and maturation, providing significant antileukemia activity [143]. Like cytarabine, vincristine also presents dosing and pharmacokinetic limitations, since it is also a cell-cycle-specific drug (which acts in the M phase). Liposomes prolong drug circulation time, promote its accumulation in tumors, and modify drug release in the tumor interstitium [144].

### 7.5. Onivyde^®^

Onivyde^®^ is another liposomal formulation recently approved (2015) by the USFDA [142]. It contains irinotecan, a semisynthetic analog of the natural alkaloid camptothecin, isolated from the stem bark of *Camptotheca acuminata*, which acts by stabilizing the complex formed by topoisomerase I and DNA, subsequently leading to DNA strand breaks and inhibition of cellular replication. Irinotecan is a prodrug that is converted by carboxylesterases into the active metabolite SN-38. Therefore, Onivyde^®^ promotes not only an increase in drug payload to the tumor—which is important because of the S-phase specificity of the drug—but also confers protection to irinotecan from premature enzymatic activation, allowing longer duration of the drug in circulation, improved biodistribution, and minimized systemic toxicity [145].

### 7.6. Vyxeos^®^

In 2017, USFDA approved Vyxeos^®^, a liposomal formulation composed of cytarabine and daunorubicin in a 5:1 molar ratio, for the treatment of different types of acute myeloid leukemia. Vyxeos^®^ provided an improved efficacy at a lower cumulative daunorubicin and cytarabine dose as compared to the free drugs combined, already used in clinical practice. The main advantage related to the implementation of these nanocarriers is the simultaneous delivery of both drugs to the target cells, which does not occur in the same proportion when these drugs are administered freely, since they exhibit different pharmacokinetic and metabolism profiles. This nanoformulation is the first approved that combines two different drugs; therefore, it could be an inspiration for other formulations [120,132].

## 8. Nanotechnology Controversies

Since nanotechnology is a relatively recent field, there are ongoing discussions on relevant aspects of nomenclature, regulation, and biological effects. Although regulatory aspects are not the focus of this review, it is worth including here a few controversies. First, “nano-words” such as “nanotechnology”, “nanoscale”, and “nanomaterials” remain undefined clearly [146]. Although relevant aspects of materials that should be considered for classification as nanotechnology-based products were provided by the USFDA (as reported at the beginning of this review), widely propagated [7] and reassured by the EMA [147], nanotechnology still lacks standardization. Several countries do not present a document discussing the attributes associated with nanoproducts, despite marketing them [148]. Thus, regulatory approval of nanomedicines is still incipient, requiring improvement and international legislation. To the best of our knowledge, the earliest guidance document regarding nanoproducts was published by the USFDA [7], ≈20 years after the approval of Doxil^®^. The most recent document was published in April 2022 [149], and it stated that nanomedicines should follow the rules already in force for drugs that do not involve nanotechnology and must, therefore, be classified in the existing categories: new drugs, biopharmaceuticals, and generics [146]. However, one question comes to mind: since nanoproducts should present distinct properties when compared with their bulk form, should the guidelines for conventional formulations be followed? Moreover, it is worth noting that even generic forms of the same nanomedicine may present distinct physicochemical properties and biological activity, such as that presented for Doxil^®^ and Lipodox^®^ [50,51], and what was observed even between Doxil^®^ and Caelyx^®^ (in theory, the same formulation) [150]. Therefore, a more cautious assessment involving bioequivalency should be conducted.

Moreover, despite the contributions of nanotechnology, some of the discussed aspects cannot be extrapolated for all cases, while other features may be sometimes overestimated. One of these factors is the role of nanopharmacology in the reversion of resistance. For example, one of the mechanisms discussed for Doxil^®^ release is through endocytosis and lysosomal processing, which can also be associated with DOX retention in the lysosomes, reducing the drug available to interact with its target [108]. Hence, it could explain the fact that, despite reducing the cardiotoxic effect of DOX, Doxil^®^ does not demonstrate superior efficacy in relation to Adriamycin^®^ [78]. Likewise, contrary to what was observed with Lipodox^®^, Abraxane^®^ use was associated with upregulation of P-gp expression in lung adenocarcinoma cell line A549 [151].

Additionally, contribution of the EPR effect on passive targeting has been questioned since more recent studies demonstrated that some factors, such as the high interstitial pressure in the tumor microenvironment and the presence of avascularized tumor areas, may reduce the contribution of EPR to drug delivery [152]. Moreover, in most of preclinical and clinical studies involving nanoformulations, including those discussed in this review, drug accumulation is compared between normal and tumor tissues, and not between conventional and nanotechnological formulations. In this context, a study performed recently described that occurrence of EPR depends on the model employed in the preclinical study: by using subcutaneous and orthotopic breast cancer models, EPR is confirmed; however, when using transgenic mouse spontaneous breast cancer models, which best mimics the patients’ conditions, its effect was negligible [153].

## 9. Conclusions and Further Perspectives

Perhaps we are still a long way from obtaining true “magic bullets”. Otherwise, nanotechnology properties have improved various outcomes compared to conventional formulations. All the difficulties in the development and approval processes of DOX and PTX nanosystems highlighted the need for thinking on formulation and nanocarrier design right at the beginning of a drug R&D process to overcome drug limitations. Moreover, when aiming to develop nanoformulations of a current drug, it is important to consider if the investment will payoff therapeutically and the new nanopharmaceutical will indeed improve drug’s pharmacokinetics or pharmacodynamics, producing benefits beyond mere tissue targeting by the EPR effect. Although the real contribution of the EPR effect upon intravenous administration in humans is still controversial, the use of alternative and local routes instead of intravenous administration of nanocarriers might facilitate active targeting, helping the nanocarrier to achieve its full therapeutic potential. Furthermore, although not frequently studied, nanotechnology might modify the pharmacodynamics properties of a drug, which is underestimated in relation to the modulation of pharmacokinetics, even though it is key when designing a novel nanocarrier. The cellular fate and recycling of nanocarrier components, and whether phospholipids and other nanocarrier components influence cell signaling and drug pharmacodynamics need to be better addressed.

## Figures and Tables

**Figure 1 pharmaceutics-14-01722-f001:**
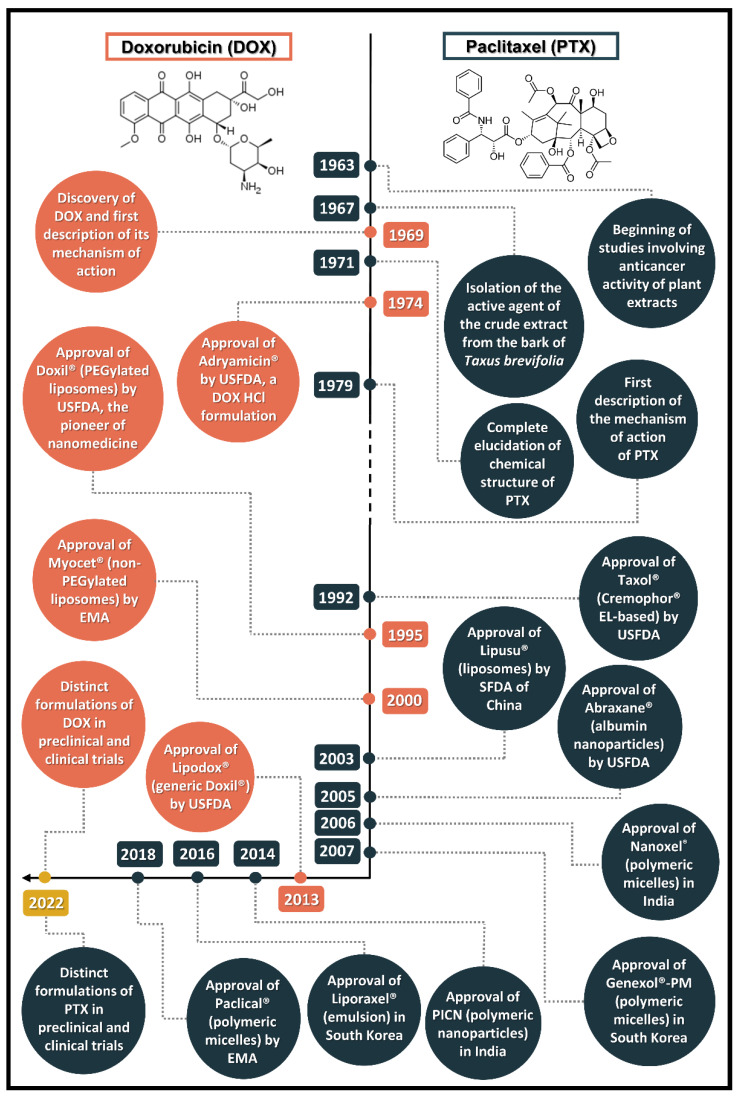
Research and development (R&D) timeline of doxorubicin (DOX) and paclitaxel (PTX), considering their conventional and nanotechnology-based dosage forms (USFDA—U.S. Food and Drug Administration; EMA—European Medicines Agency; SFDA—State Food and Drug Administration).

**Figure 2 pharmaceutics-14-01722-f002:**
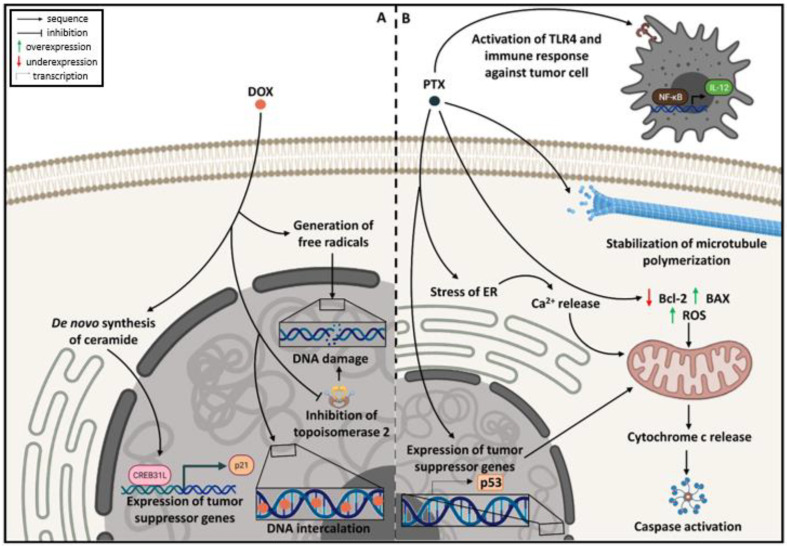
Mechanisms of action of doxorubicin—DOX (**A**) and paclitaxel—PTX (**B**); both drugs exert their effects by hampering tumor cell physiopathology. PTX also enhances patient’s immune response. (ER = endoplasmic reticulum; ROS = reactive oxygen species) (Created with BioRender.com).

**Figure 3 pharmaceutics-14-01722-f003:**
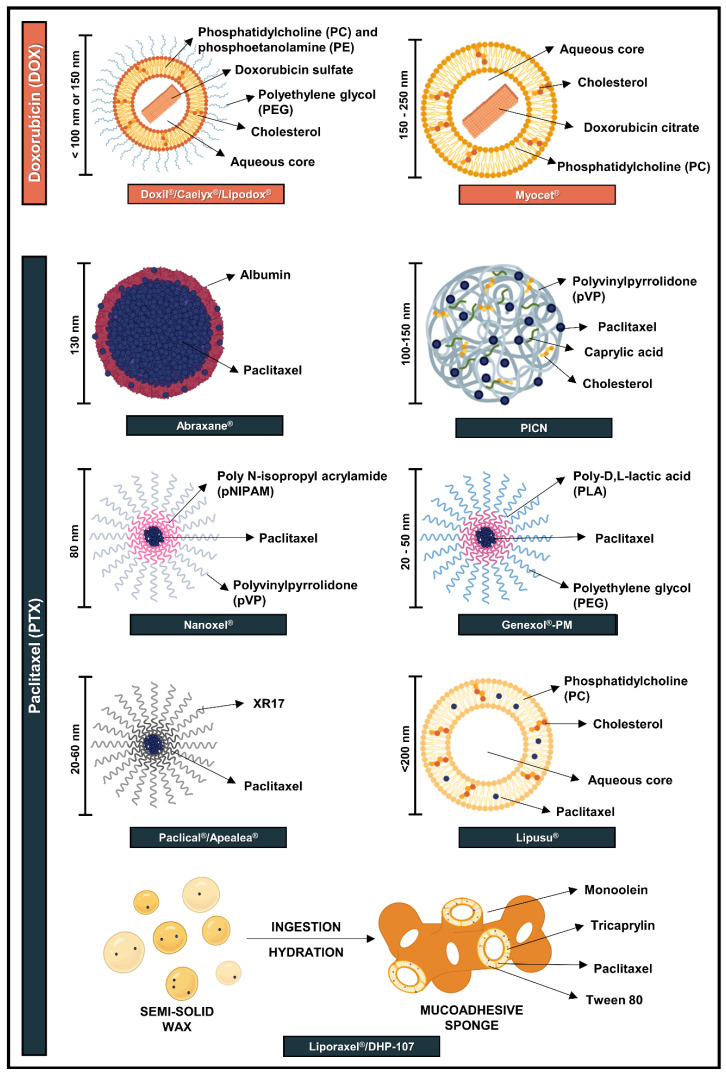
Approved nanoformulations employing doxorubicin (DOX) or paclitaxel (PTX) and their respective compositions. For DOX, there are two types of liposomes: pegylated and non-pegylated, whereas PTX presents a wider variety of nanocarriers: polymeric nanoparticles, polymeric micelles, and lipid-based formulations. (XR17 = isoforms of N-retinoyl-cysteic acid methyl esters) (Created with BioRender.com).

**Figure 4 pharmaceutics-14-01722-f004:**
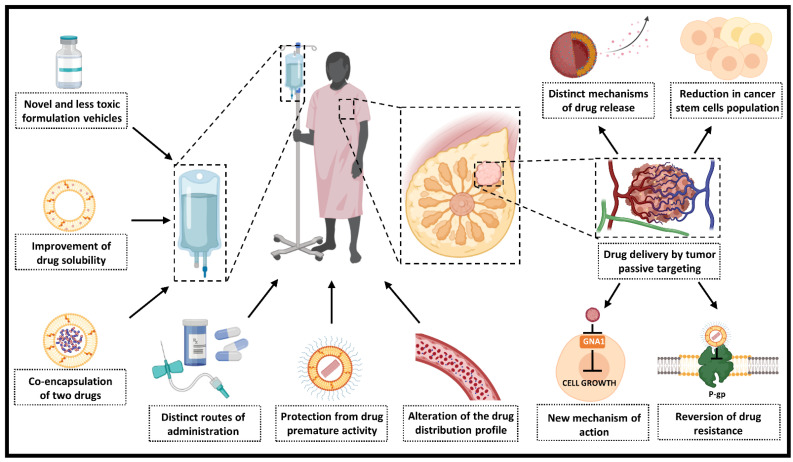
Some contributions of nanotechnology for cancer treatment. This illustration summarizes the contributions from the formulation step to the tumor cells and higher efficacy. Thus, this highlights that nanotechnology is more than just formulation and investments in this strategy for drug delivery are worth it (Pgp: P-glycoprotein) (Created with BioRender.com).

**Figure 5 pharmaceutics-14-01722-f005:**
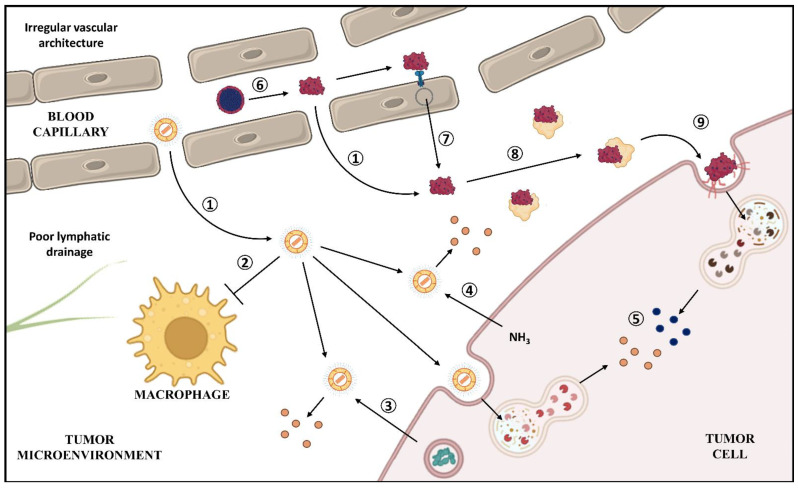
Proposed mechanisms for Doxil^®^ and Abraxane^®^ tumor accumulation. After being injected intravenously, the pegylated liposomes of doxorubicin (DOX—orange) formulation may reach the tumor through passive targeting/enhanced permeation-retention (EPR) effect (1), since this site presents irregular vascular architecture with fenestrated capillaries and impaired lymphatic drainage. The presence of polyethylene glycol (PEG) in Doxil^®^ corona provides a hydration layer that repels opsonins from the liposome and avoids phagocytosis by cells of the reticuloendothelial system (RES), such as macrophages (2). DOX may be released through three distinct mechanisms: secretion of phospholipases (3) or ammonia (4) by tumor cells or endocytosis followed by lysosomal processing (5). Abraxane^®^ presents other peculiarities: it probably dissociates into smaller complexes composed of albumin and paclitaxel (PTX; blue) (6). These structures, such as Doxil^®^, may accumulate in tumor microenvironment, by EPR effect (1), transcytosis through binding receptors (glycoprotein 60—gp60) in endothelial cells (7), and interaction with secreted protein acidic and rich in cysteine (SPARC—yellow) (8). Albumin may be endocytosed through the caveolin-1 pathway (9), followed by lysosomal processing with subsequent drug release (5). (Created with BioRender.com).

**Figure 6 pharmaceutics-14-01722-f006:**
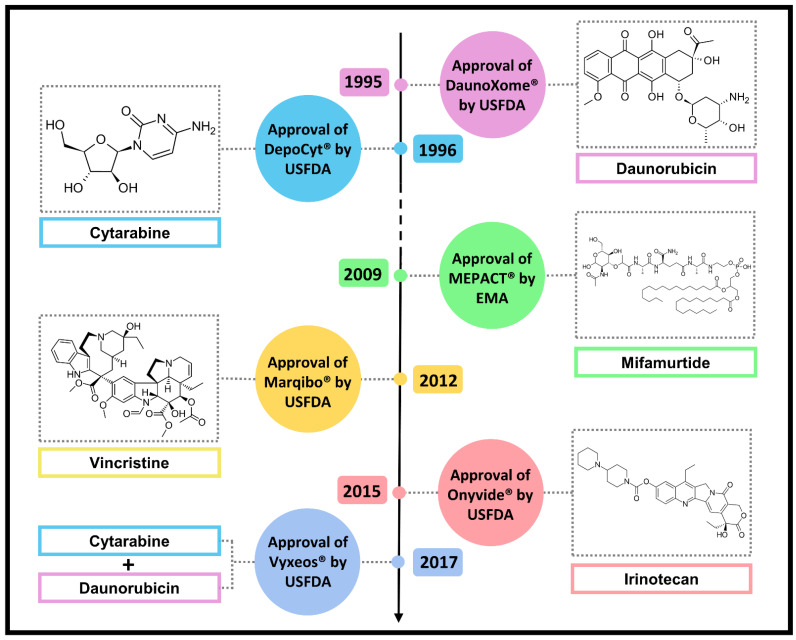
Timeline for approval of other natural-product-derived anticancer drugs, considering their nanotechnology-based dosage forms (USFDA—U.S. Food and Drug Administration; EMA—European Medicines Agency).

**Table 1 pharmaceutics-14-01722-t001:** Main characteristics of doxorubicin (DOX) and paclitaxel (PTX) formulations, such as their administration route, therapeutic applications, recommended dose, and maximum tolerated dose (MTD). (* indicates cumulative dose) (NSCLC—non-small cell lung cancer).

Drug	Formulation	Administration Route	Therapeutic Applications	Recommended Dose	Maximum Tolerated Dose (MTD)
DOX	Adriamycin^®^	Intravenous infusion	A wide variety of tumors (hematologic, solid, and neural tumors)	40–75 mg/m^2^	* 500 mg/m²
	Doxil^®^/Caelyx^®^/Lipodox^®^		AIDS-related Kaposi’s sarcoma, multiple myeloma, and ovarian and breast cancers	20–50 mg/m^2^	120 mg/m^2^
	Myocet^®^		Metastatic breast cancer	60–75 mg/m^2^	75–135 mg/m^2^
PTX	Taxol^®^	Intravenous infusion	Ovarian and breast cancers	135–175 mg/m^2^	240 mg/m^2^
	Abraxane^®^		Breast and pancreas cancers and NSCLC	260 mg/m^2^	300 mg/m^2^
	PICN		Breast cancer	260 mg/m^2^	325 mg/m^2^
	Genexol^®^-PM		Breast and pancreas cancers, NSCLC, AIDS-related Kaposi’s sarcoma	300–390 mg/m^2^	390 mg/m^2^
	Nanoxel^®^		Breast cancer and NSCLC	330 mg/m^2^	375 mg/m^2^
	Paclical^®^/Apealea^®^		Ovarian cancer	250 mg/m^2^	250 mg/m^2^
	Lipusu^®^		NSCLC, ovarian and breast cancers	175 mg/m^2^	no data
	Liporaxel^®^	Oral administration	Gastric cancer	200 mg/m^2^	600 mg/m^2^

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
