# Peer review of "Beyond Formulation: Contributions of Nanotechnology for Translation of Anticancer Natural Products into New Drugs"

_pharmaceutics, 2022, doi:10.3390/pharmaceutics14081722_

Round 1

Reviewer 1 Report

This work consists of analyzing the contributions made by various authors in the field of nanotechnology and its application to cancer. Two of the chemical compounds used in chemotherapy, doxorubicin (DOX) and paclitaxel (PTX), are analyzed. 

The authors describe chronologically from the beginning of the use of these drugs in a traditional way to the formulations that are currently in use, each of their formulations, their therapeutic concentrations, and their adverse effects to use them, and their advantages and disadvantages. They also analyzed the cost-benefit advantage. 

As Doxil® and  Abraxane® are the most studied nanoformulations of DOX and PTX, respectively, the  contribution of these formulations to pharmacokinetics and pharmacodynamics of these  drugs is discussed in detail.

Suggestion: the authors could also analyze the lactoferrin-coated or Lf-conjugated carriers to delivery these drugs and their use in nanoapplications, to have a better manuscript.

Author Response

We thank the Reviewers for their constructive comments and suggestions in the manuscript. In this document, we respond to their comments in a point-by-point fashion.

  1. This work consists of analyzing the contributions made by various authors in the field of nanotechnology and its application to cancer. Two of the chemical compounds used in chemotherapy, doxorubicin (DOX) and paclitaxel (PTX), are analyzed. The authors describe chronologically from the beginning of the use of these drugs in a traditional way to the formulations that are currently in use, each of their formulations, their therapeutic concentrations, and their adverse effects to use them, and their advantages and disadvantages. They also analyzed the cost-benefit advantage. As Doxil® and Abraxane® are the most studied nanoformulations of DOX and PTX, respectively, the contribution of these formulations to pharmacokinetics and pharmacodynamics of these  drugs is discussed in detail.

Suggestion: the authors could also analyze the lactoferrin-coated or Lf-conjugated carriers to delivery these drugs and their use in nanoapplications, to have a better manuscript.

Answer: A new subsection to discuss potential contributions of nanotechnology based on formulations on earlier stages of development was included in the manuscript. In this session, the use of active targeting and lactoferrin-coated nanocarriers was discussed (Session “Other possible contributions of nanotechnology for DOX and PTX”, p. 18). We also included other potential contributions, such as the possibility to co-encapsulate drugs, to increase drug solubility, and thermally induced drug release.

Reviewer 2 Report

The entitled review manuscript "Beyond formulation: contributions of nanotechnology for translation of anticancer natural products into new drugs" from Miguel et al involved the nanotechnology contribution for the development of several natural products, mainly DOX and PT clinical systems.

Besides, the manuscript is organized, easy to follow and it covers in detail all the aspects the authors specify in the abstract. Furthermore, it is an interesting topic to cover due to the importance of novel nanoformulations in pharmacology and biomedicine fields of research/clinical applications.

Additionally, the authors used appropriate images, figures, schemes, and tables to show the updated bibliographical analyses they performed.

According to this, I suggest improving the quality of Figure 4 (it is not possible to read the words related to each image at 100% page).

Moreover, (figure 6) all the words inside the colored circles are not clear enough for the reader (they look blurry), maybe using black colour (for words) could fix this problem.

It is important to note that the authors also included one item related to the controversies regarding the use of nanotechnology in pharmaceutical formulations. This point demonstrates the critical point of view the authors had about the information and the clinical utilization of these nanosystems.

In this topic, it could be important to add information about the FDA and EMA regulations for these kinds of nanoformulations to improve the discussion and the conclusions.

I would like to invite the authors to add the abbreviation list of words at the end of this manuscript.

Finally, I recommend the acceptance of this manuscript after the authors performed the suggested corrections/additions.

Author Response

We thank the Reviewers for their constructive comments and suggestions in the manuscript. In this document, we respond to their comments in a point-by-point fashion.

The entitled review manuscript "Beyond formulation: contributions of nanotechnology for translation of anticancer natural products into new drugs" from Miguel et al involved the nanotechnology contribution for the development of several natural products, mainly DOX and PT clinical systems.

Besides, the manuscript is organized, easy to follow and it covers in detail all the aspects the authors specify in the abstract. Furthermore, it is an interesting topic to cover due to the importance of novel nanoformulations in pharmacology and biomedicine fields of research/clinical applications. Additionally, the authors used appropriate images, figures, schemes, and tables to show the updated bibliographical analyses they performed.

  1. According to this, I suggest improving the quality of Figure 4 (it is not possible to read the words related to each image at 100% page).

      Answer: The Figure resolution was improved.

  1. Moreover, (figure 6) all the words inside the colored circles are not clear enough for the reader (they look blurry), maybe using black colour (for words) could fix this problem.

        Answer: The font color was modified.

  1. It is important to note that the authors also included one item related to the controversies regarding the use of nanotechnology in pharmaceutical formulations. This point demonstrates the critical point of view the authors had about the information and the clinical utilization of these nanosystems. In this topic, it could be important to add information about the FDA and EMA regulations for these kinds of nanoformulations to improve the discussion and the conclusions.

          Answer: We included few controversies regarding definitions and regulatory aspects of nanotechnology-based products (p. 23).

  1. I would like to invite the authors to add the abbreviation list of words at the end of this manuscript.

         Answer:  A list of abbreviation was included at the end of the manuscript.

  1. Finally, I recommend the acceptance of this manuscript after the authors performed the suggested corrections/additions.

        Answer: We appreciate this recommendation, and hope that the modifications performed improved the manuscript.